# Spinal cord injury and risk of overall and type specific cardiovascular diseases: A meta-analysis

**ShengZhong Luo**[1,2,3], **Tianlong Wu**[1,2,3], **Xigao Cheng**[1,2,3]*

**1** Department of Orthopaedics, The Second Affiliated Hospital, Jiangxi Medical College, Nanchang University, Nanchang, Jiangxi, China, **2** Institute of Orthopedics of Jiangxi Province, Nanchang, Jiangxi, China, **3** Institute of Minimally Invasive Orthopedics, Nanchang University, Nanchang, Jiangxi, China

* xigaocheng@hotmail.com

**Data Availability Statement:** All relevant data are within the manuscript and its Supporting information files.

**Funding:** The author(s) received no specific funding for this work.

## Abstract

### Background

Cardiovascular disease (CVD) is a growing concern among people with spinal cord injury (SCI). This meta-analysis aims to explore the risk of overall CVD and specific types of cardiovascular events among SCI patients.

### Methods

This meta-analysis is registered on PROSPERO (CRD CRD42024537888). The data sources comprised PubMed, Embase, the Cochrane Library, and reference lists of the included studies. The literature collection span is from database establishment until April 17, 2024. This meta-analysis encompassed observational studies investigating the association between SCI and the risk of overall types of CVD or specific CVD types. Risk of bias was evaluated utilizing the Newcastle-Ottawa Quality Assessment Scale (NOS) and the Agency for Healthcare Research and Quality (AHRQ) Scale. Odds ratios (ORs) with 95% confidence intervals (CIs) were aggregated using a random-effects model.

### Results

Our initial search generated 5357 relevant records form these international databases. This meta-analysis encompassed 9 observational studies involving 2,282,691 individuals, comprising 193,045 patients with SCI and 2,209,646 controls. We observed a 1.56-fold [OR = 1.56, 95% CI (1.43, 1.70), $I^2$ = 91.3%, P < 0.001] rise in the risk of overall types of CVD among SCI patients, with a 1.82-fold increase in males and a 1.76-fold increase in females. SCI patients without comorbidities exhibited a 2.10-fold elevated risk of overall CVD types, while those with comorbidities had a 1.48-fold increased risk. Concerning specific CVD types, SCI patients showed a 1.58-fold [OR = 1.57, 95% CI (1.22, 2.03), $I^2$ = 92.4%] higher risk of myocardial infarction, a 1.52-fold [OR = 1.52, 95% CI (1.07, 2.16), $I^2$ = 88.7%] increase in atrial fibrillation, a 1.64-fold [OR = 1.64, 95% CI (1.22, 2.20), $I^2$ = 95.5%] elevation in heart failure risk, and 2.38-fold [OR = 2.38, 95% CI (1.29, 4.40), $I^2$ = 92.5%]

**Competing interests:** The authors have declared that no competing interests exist.

increments in stroke risk. But there was no statistically significant difference in the risk of hypertension [OR = 1.54, 95% CI (0.98, 2.42), $I^2$ = 96.6%].

## Conclusions

The risk of overall CVD in SCI patients surpassed that of the non-SCI control group, with elevated risks of specific cardiovascular events like myocardial infarction, atrial fibrillation, heart failure, and stroke. Clinicians should prioritize awareness of CVD risks in SCI patients.

## 1. Introduction

Spinal cord injury (SCI) results from damage to the spinal cord, causing various degrees of sensory, motor, and autonomic dysfunction [1–3]. It constitutes a significant health burden due to premature mortality and persistent disability. According to estimates by Global Burden of Disease (GBD) SCI collaborators, over 20 million people worldwide were affected by SCI in 2019, with around 900,000 new cases reported. Between 1990 and 2019, SCI prevalence rose by 81.5%, while the incidence increased by 52.7%. SCI profoundly impacts disability rates and overall well-being [3]. SCI often triggers secondary health complications like pressure ulcers, urinary tract infections, and respiratory issues, exacerbating the challenges to quality of life [4]. In addition, evidence shows that SCI is also associated with prostate cancer [5], dysphagia [6], cardiometabolic disease and increased risk of venous thrombosis [7], especially the risk of cardiovascular disease (CVD), which deserves further attention [8].

CVD is the leading cause of death in the population [9], and SCI patients appear to develop CVD earlier compared to the general population [10]. Patients with SCI bear a heavy burden of traditional CVD risk factors, including dyslipidemia and diabetes, and exhibit anatomical, metabolic, and physiological changes, as well as significant reductions in physical activity after injury [11, 12]. Hence, it is imperative for primary care physicians and cardiologists to recognize the criticality of promptly diagnosing and managing CVD risk among individuals with SCI. Yet knowledge about specific.

CVD such as hypertension, heart failure and myocardial infarction in SCI patients is still largely limited. To a large extent, there is still a lack of systematic evidence on the relationship between all-cause or specific cardiovascular events. Therefore, a systematic review and meta-analysis of existing evidence from observational studies were conducted to quantify the overall and specific types of CVD risk in SCI.

## 2. Methods

This meta-analysis follows the updated guidance and exemplars for reporting systematic reviews [13]. Our protocol has been registered on International prospective register of systematic reviews (PROSPERO) under the registration number (CRD42024537888).

### 2.1 Data sources

We conducted comprehensive searches of PubMed, EMBASE, and the Cochrane Library from their inception until April 17, 2024, without restrictions. We used subject terms (Emtree in Embase, MeSH in PubMed and Cochrane Library) and corresponding keywords focused on spinal cord injury and various cardiovascular conditions including heart diseases, coronary artery disease, myocardial infarction, heart failure, angina pectoris, stroke, and hypertension.

Our search strategy, modeled on previous high-quality meta-analyses [14–16], included examining reference lists of retrieved studies for additional relevant articles. The complete search strategy is detailed in **S1-S3 Tables in** S1 File.

## 2.2 Study selection

The initial records retrieved were imported into NoteExpress reference management software, where duplicate records were identified and removed. Two authors (ZS Luo and TL Wu) independently assessed titles and abstracts to exclude irrelevant records and categorized the remaining records as either inclusion, exclusion, or uncertain. For uncertain records, full texts were reviewed to confirm eligibility for inclusion. Any disagreements were resolved through group discussion.

## 2.3 Eligibility criteria

We considered studies meeting the following criteria eligible for inclusion: observational studies employing cohort, case-control, or cross-sectional designs; participants with a documented history of SCI and at least one reported CVD; presentation of point estimates (risk ratios [RR] or odds ratios [OR], incidence rate ratios [IRR]) with corresponding 95% confidence intervals (CI). Studies lacking pertinent data on qualified control groups or CVD outcomes would be excluded. Additionally, publications lacking original data, such as comments, editorials, meeting abstract and reviews would be excluded. In cases of overlapping datasets, we prioritized studies with the largest sample size or the most comprehensive data for analysis.

## 2.4 Data extraction

We created a data extraction template using Excel software (Microsoft Corporation, USA). Two authors (ZS Luo and TL Wu) independently extracted information from eligible cohorts. The extracted data included details such as first author, publication date, country, event numbers, exposure numbers, confounders, and effect size. Data extraction was cross-validated, and any discrepancies were resolved through discussion with third reviewer (XG Cheng).

## 2.5 Study quality

For cohort and case control studies, the Newcastle-Ottawa Quality Assessment Scale (NOS) [17] (Available from: http://www.ohri.ca/programs/clinical_epidemiology/oxford.asp) was employed to assess the quality of included studies across three domains: selection, comparability, and outcomes. Cohort and case-control studies were assigned scores ranging from 0 to 9 stars, with higher scores denoting superior study quality. NOS ratings of $\geq$7, 4–6, and 0–3 were categorized as high, moderate, and low quality, respectively. For cross-sectional studies, the Agency for Healthcare Research and Quality (AHRQ) [18] utilized an 11-item assessment, where a score of "yes" equated to 1 point and "no" or "unclear" to 0 points. Scores falling within the ranges of 0–3, 4–7, and 8–11 were designated as low, medium, and high quality, respectively.

## 2.6 Data synthesis

The extracted meta-analysis data underwent analysis using STATA statistical software version 14.0. A bilateral P-value with an $\alpha$ level of 0.05 was deemed significant. Employing the random effects model (DerSimonian and Laird methods), we computed the combined OR with a 95% CI to ascertain the association between SCI and CVD risk. Given the relatively low incidence of SCI and CVD, the OR was utilized as a proxy for the RR. We prioritized risk estimation

from multivariate models that fully adjusted for confounding factors. For studies reporting risk estimates solely for specific CVD types without comprehensive data on overall CVD, we utilized random effects models based on heterogeneity levels to synthesize specific risk estimates. Heterogeneity across studies was quantified using $I^2$ statistics (0–25%: low heterogeneity, 25%-50%: moderate heterogeneity, 50%-75%: substantial heterogeneity, 75%-100%: high heterogeneity). We conducted subgroup analysis based on specific types of CVD, gender, and the presence of comorbidities [19].

## 3 Results

### 3.1 Study selection

Our initial search generated 5357 relevant records, of which 1120 duplicates were removed. Subsequently, 4211 records were excluded after screening titles and abstracts for relevance to our topic. The remaining 26 studies underwent further scrutiny for eligibility. Ultimately, 9 observational studies [11, 20–27] met the inclusion criteria for the meta-analysis. The flowchart depicting the study selection process is presented in Fig 1.

### 3.2 Study characteristics

Nine studies encompassing 2,282,691 individuals (193,045 with SCI and 2,089,646 controls) were included in this analysis. The study populations predominantly originated from Taiwan Province in China (*n* = 4), the United States (*n* = 2), Canada (*n* = 2) and Korea (*n* = 1). These studies were published between 2012 and 2024, with 7 cohort studies [11, 20, 21, 23, 24, 26, 27] and 2 cross-sectional studies [22, 25]. The analysis focused on six specific types of CVD: cardiometabolic morbidity, cardiac dysrhythmias, heart failure, myocardial infarction, atrial fibrillation, and acute coronary syndrome. Adjusted confounders varied slightly among the included studies, with age, sex, and education being the most commonly adjusted variables. Table 1 provides a summary of the characteristics of the included studies.

### 3.3 Quality assessment

All 7 cohort studies [11, 20, 21, 23, 24, 26, 27] had scores ≥ 7, indicating the high quality of the cohort studies included in this meta-analysis. The scores of the two cross-sectional studies [22, 25] were 7 and 8 respectively, which were of medium quality. Table 1 summarizes the scores of the risk of bias.

### 3.4 SCI and risk of overall type of CVD

A total of 8 cohorts examined the risk of overall types of CVD in SCI patients. The pooled analysis indicated a heightened risk of overall types of CVD among SCI patients [OR = 1.56, 95% CI (1.43, 1.70), I2 = 91.3%, P < 0.001; Fig 2]. Given the significant heterogeneity, we conducted sensitivity analysis by systematically removing each study to identify potential sources of heterogeneity. Results revealed that the findings remained consistent and robust across all iterations, indicating the stability of our meta-analysis results (S1 Fig in S1 File).

### 3.5 SCI and risk of specific type of CVD

Six studies evaluated the risk of specific types of CVD in SCI patients. The pooled analysis revealed an elevated risk of myocardial infarction [OR = 1.57, 95% CI (1.22, 2.03), $I^2$ = 92.4%, Fig 3], atrial fibrillation [OR = 1.52, 95% CI (1.07, 2.16), $I^2$ = 88.7%, Fig 3], heart failure [OR = 1.64, 95% CI (1.22, 2.20), $I^2$ = 95.5%, Fig 3], and stroke [OR = 2.38, 95% CI (1.29, 4.40),

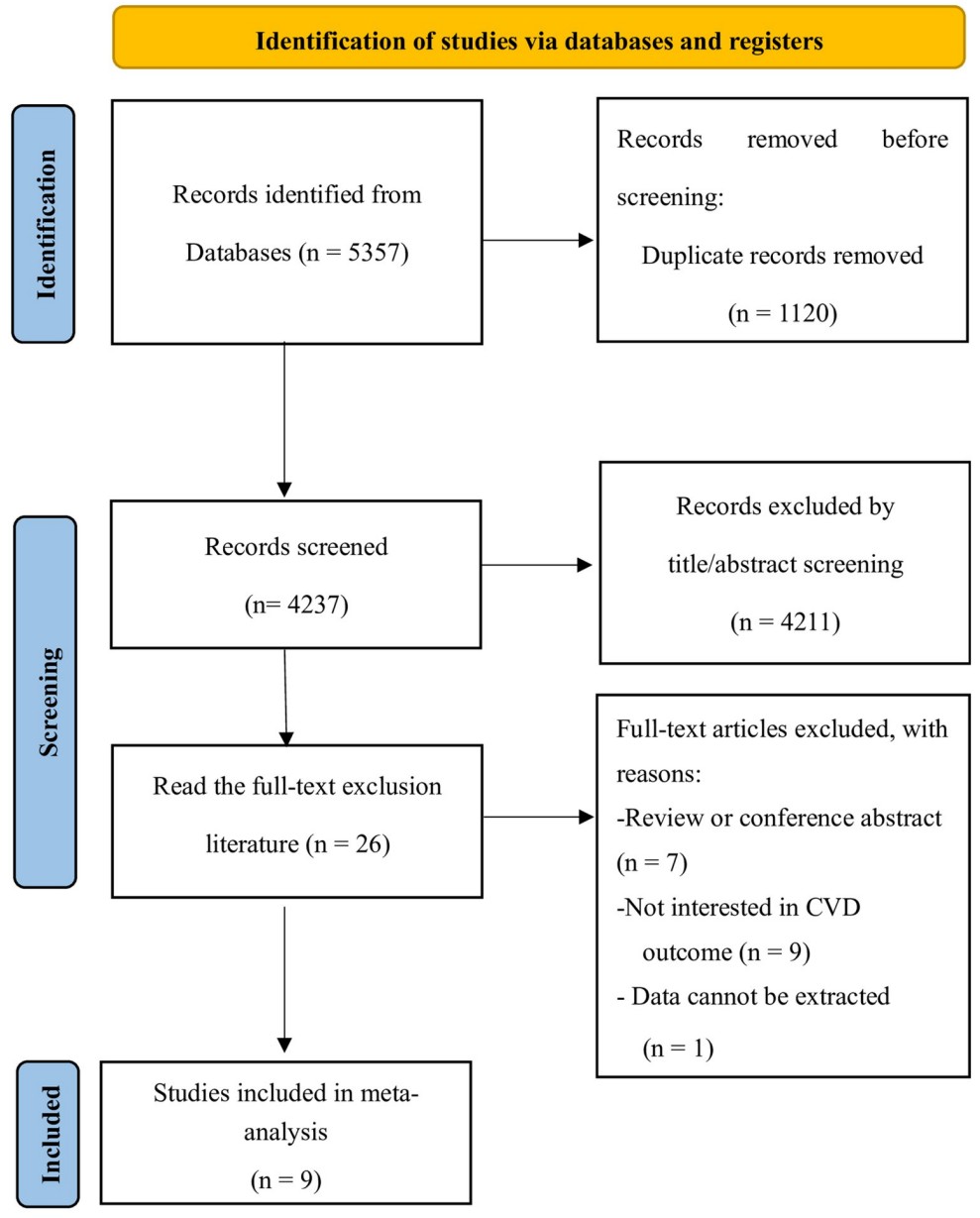

**Fig 1. Flow chart of literature screening.**

$I^2$ = 92.5%, Fig 3] among SCI patients. However, there was no statistically significant difference in the risk of hypertension [OR = 1.54, 95% CI (0.98, 2.42), $I^2$ = 96.6%, Fig 3].

### 3.6 Subgroup analysis

**Gender.** The risk of overall type of CVD in SCI patients increases across different genders, with males [OR = 1.82, 95% CI (1.43, 2.32), $I^2$ = 88.0%, Fig 4], and females [OR = 1.76, 95% CI (1.28, 2.43), $I^2$ = 89.9%, Fig 4].

**Comorbidities.** The risk of CVD increases in SCI patients with or without comorbidities, with no comorbidities [OR = 2.10, 95% CI (1.44, 3.07), $I^2$ = 93.0%, Fig 5], and comorbidities [OR = 1.48, 95% CI (1.28, 1.72), $I^2$

**Table 1. Characteristics of included studies.**

| Author | Year | Country | Design | Data source | Enroll period | Follow-up period | Diagnoses | CVD events | SCI cases | Non-SCI Controls | Confounders adjusted | Effect size | NOS or AHRQ score |
|---|---|---|---|---|---|---|---|---|---|---|---|---|---|
| Yoo | 2024 | Korea | Cohort study | The National Health Insurance Service (NHIS) | 2010–2018 | 4.3±2.6 years | Diseases-10th revision (ICD-10) | MI = 169, HF = 426, AF = 158 | 5,083 | 19,320 | Age, sex, and Charlson comorbidity index, socioeconomic position, smoking, alcohol consumption, physical activity, and comorbidities | RR | 9 |
| Michelle | 2023 | USA | Cohort study | Clinformatics DataMart Database | 2007–2017 | 4 years | ICD-9-CM or ICD-10-CM | CVD = 628, HF = 525, HP = 742, MI = 163, Stroke = 299 | 9,081 | 24,074 | Sex, education level, net worth, and Elixhauser comorbidities. | RR | 7 |
| Jia | 2023 | Canada | Cross-sectional study | Canadian Community Health Survey | / | / | Self-reported | HD = 112 | 354 | 60,605 | Age, smoking status, diabetes status, physical activity levels, alcohol consumption, and education level | OR | 8 |
| Peterson | 2021 | USA | Cohort study | Clinformatics DataMart Database | 2001–2017 | / | ICD-9-CM | CM = 1,527, CD = 2,477, HF = 1,393, HP = 1,636 | 9,081 | 1,474,232 | Age, sex, race, geographic region, modified exhauster comorbidity index, education, income | RR | 8 |
| Wang | 2016 | China (Taiwan) | Cohort study | National Health Insurance Research Database | 2000–2011 | 5.69 years | ICD-9-CM | AF = 2,045 | 41,691 | 166,724 | Age, sex, and the comorbidities of diabetes, hypertension, hyperlipidemia, COPD, heart failure, CAD, stroke, hyperthyroidism, cancer, and chemotherapy | RR | 8 |
| Yang | 2015 | China (Taiwan) | Cohort study | National Health Insurance Research Database | / | / | ICD-9-CM | MI = 1,171 | 22,197 | 88,788 | Age, sex, and comorbidity | RR | 7 |
| Cragg | 2013 | Canada | Cross-sectional study | Canadian Community Health Survey | 2010 | / | Self-reported | HD = 354, Stroke = 356 | 61,031 | 60,959 | Sex and age | OR | 7 |
| Chen | 2013 | China (Taiwan) | Cohort study | Taiwan National Health Insurance claim data | 2000–2009 | 10 years | ICD-9-CM | ACS = 1,504 | 41,721 | 166,884 | Age, sex, diabetes, hypertension and hyperlipidemia | RR | 7 |
| Wu | 2012 | China (Taiwan) | Cohort study | Taiwan's National Health Insurance Research Database | 1998–2002 | 4 years | ICD-9-CM | Stroke = 292 | 2,806 | 28,060 | Age, sex | RR | 7 |

ICD: International Classification of Diseases; CM: Cardiometabolic morbidity; CD: Cardiac dysrhythmias; HF: Heart failure; MI: Myocardial infarction; AF: Atrial fibrillation; CVD: Cardiovascular disease; HD: Heart disease; ACS: Acute coronary syndrome; RR: risk ratios; OR: odds ratios.

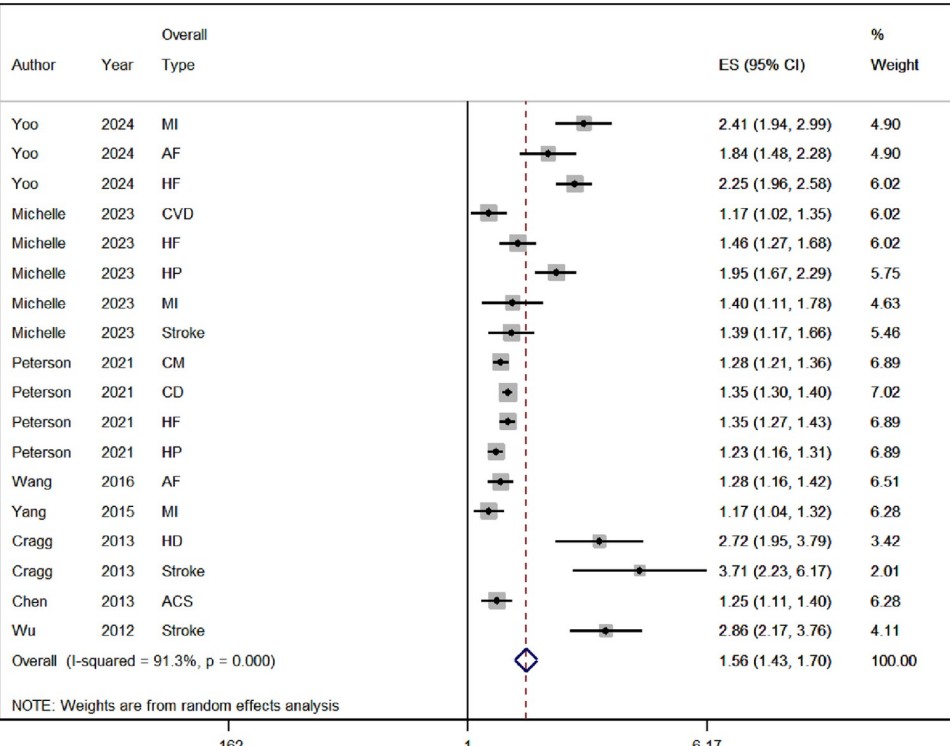

ES: effect size.

**Fig 2. Forest plot for the risk of overall type of CVD in SCI.**

## 4. Discussion

### Main findings

This meta-analysis encompassed 9 observational studies involving 2,282,691 individuals, comprising 193,045 patients with SCI and 2,209,646 controls. We observed a 1.56-fold rise in the risk of overall types of CVD among SCI patients, with a 1.82-fold increase in males and a 1.76-fold increase in females. Additionally, SCI patients without comorbidities exhibited a 2.10-fold elevated risk of overall CVD types, while those with comorbidities had a 1.48-fold increased risk. Concerning specific CVD types, SCI patients showed a 1.58-fold higher risk of myocardial infarction, a 1.52-fold increase in atrial fibrillation, a 1.64-fold elevation in heart failure risk, and 2.38-fold increments in stroke risk.

### Interpretation of findings

Determining CVD risk in SCI patients poses a significant challenge due to various factors at play [8]. Sensory and autonomic dysfunction post-SCI can obscure typical CVD symptoms, potentially leading to undetected or late-stage diagnoses, often resulting in fatal outcomes, hence explaining CVD's prominence as a cause of death in this population [28–30]. Limited mobility in SCI patients compounds the issue, hindering access to preventive care and treatment [31]. While large-scale studies on CVD risk in SCI patients are lacking, there's strategic importance in longitudinally collecting risk factors and assessing CVD event risk. Our meta-analysis revealed a consistent elevation in overall CVD risk among SCI patients across various

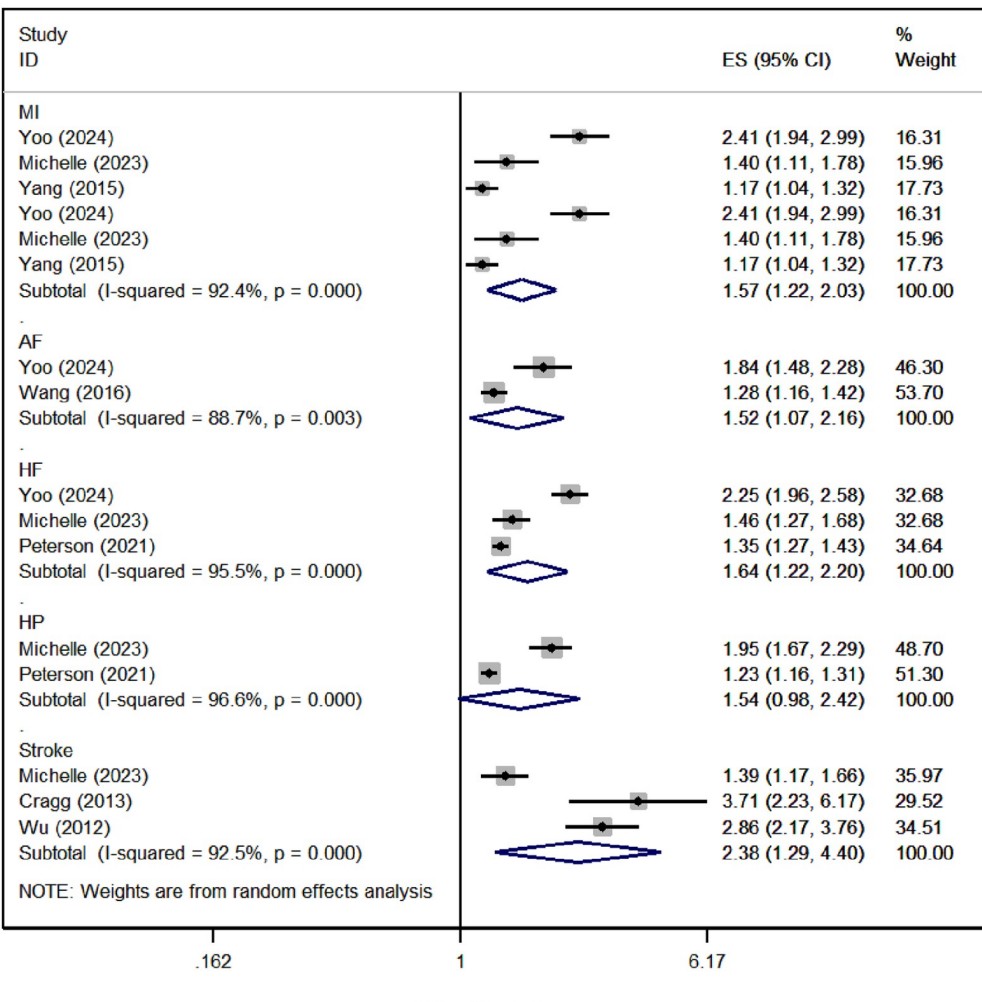

**Fig 3. Forest plot for the risk of specific type of CVD in SCI.**

cohorts and cross-sectional studies, irrespective of gender or comorbidity status. Intriguingly, SCI patients with comorbidities exhibited a lower overall risk of CVD events compared to those without comorbidities. We attribute this finding to comorbidities prompting earlier and more extensive healthcare interventions among SCI patients, thus mitigating CVD risk at earlier stages. Furthermore, our analysis revealed notable elevations in specific cardiovascular event risks among SCI patients: a 1.58-fold increase in myocardial infarction risk, a 1.52-fold increase in atrial fibrillation risk, a 1.64-fold increase in heart failure risk, and a substantial 2.38-fold increase in stroke risk. Notably, stroke risk was the highest among SCI patients, with myocardial infarction ranking second. These findings furnish clinicians with contemporary evidence to promptly identify and prioritize attention towards these specific cardiovascular events.

Currently, two predominant viewpoints exist regarding the specific mechanisms driving CVD risk in SCI patients: chronic inflammation and neurogenic etiology [8]. SCI triggers a persistent inflammatory response within the spinal cord, characterized by a smoldering cascade of inflammation. This process stimulates the activation of neutrophils and macrophages,

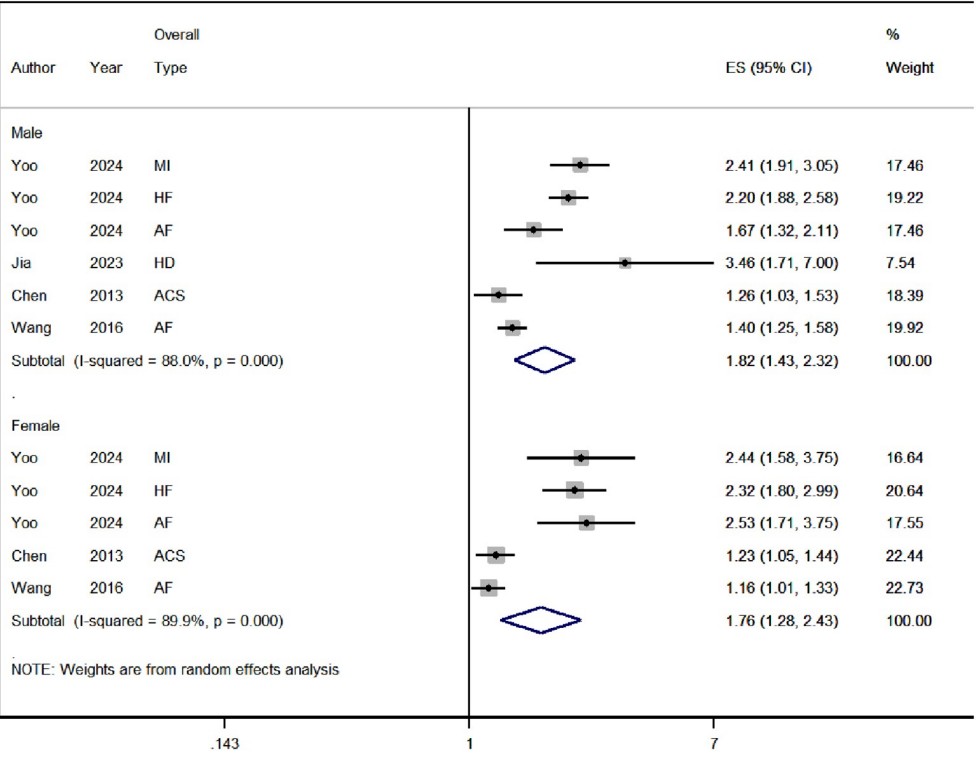

ES: effect size.

**Fig 4. Forest plot of overall type risk of CVD by gender in SCI.**

along with the production of autoreactive lymphocytes and autoantibodies. Furthermore, levels of C-reactive protein and interleukin-6 continue to rise post-injury, exacerbated by the loss of innervation in the lymphatic and endocrine systems below the injury site. Additionally, the accumulation of abdominal adipose tissue post-injury contributes to systemic inflammation by secreting pro-inflammatory adipokines and cytokines [32–34]. SCI patients often lose sympathetic nervous system activity, heart rate, and blood pressure, leading to reduced cardioprotective effects, which may increase the risk of cardiovascular events. In addition, spinal cord injury can also affect the parasympathetic nervous system, causing remodeling of vague nerve afferent fibers, resulting in reduced vague nerve sensory processing and ultimately leading to neurogenic obesity [35].

## Strengths and limitations

This meta-analysis comprehensively explores the relationship between SCI patients and the risk of overall and specific types of CVD, providing the best evidence-based approach for emphasizing the prevention of CVD risk in SCI patients. We acknowledge several limitations in our study. Firstly, as with all meta-analyses of observational studies, our analysis is constrained by the variability in study types and quality. Observational studies are susceptible to recall bias and inflated associations, typically yielding low-quality evidence. Secondly, our study exhibits some heterogeneity. While sensitivity analysis indicates robust results, clinical and methodological differences inevitably contribute to variability. We employed random-

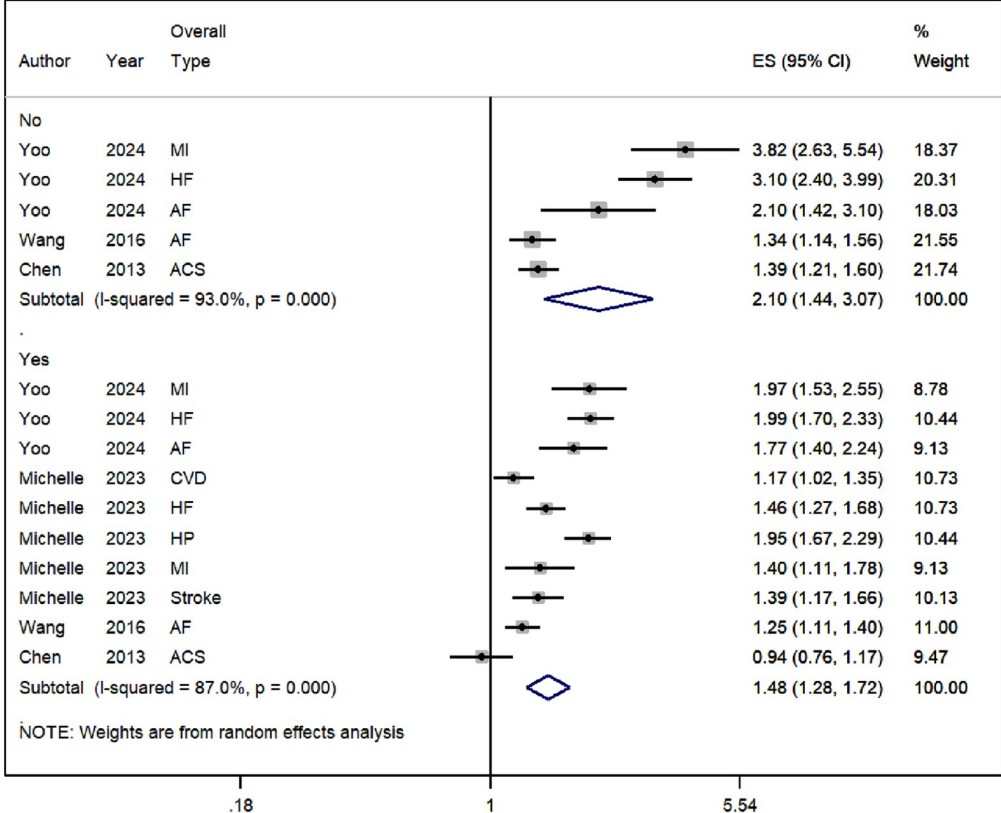

ES: effect size.

**Fig 5. Forest plot of overall type risk of CVD by comorbidities in SCI.**

effects models to mitigate heterogeneity's impact on outcomes. Despite this, our findings offer statistically reliable evidence. Thirdly, due to limited data availability, we did not conduct meta-analyses for certain CVD types, as only 1–2 studies provided effect sizes and estimates. We anticipate future updates and higher-quality research to facilitate comprehensive meta-analyses for specific CVD associations.

## 5. Conclusions

The risk of overall types of CVD in SCI patients was significantly higher than that in the non-SCI control group, and the increased risk of specific types of cardiovascular events such as myocardial infarction, atrial fibrillation, heart failure, and stroke was also associated. Clinicians must be aware of the importance of the potential risks of CVD in SCI patients.

## Supporting information

**S1 File. 1009-Supplementary.**
(DOC)

**S1 Data. Data submission.**
(XLS)

**S1 Checklist. PRISMA 2020 checklist.**
(DOCX)

## Author Contributions

**Conceptualization:** ShengZhong Luo, Xigao Cheng.

**Data curation:** ShengZhong Luo, Tianlong Wu.

**Formal analysis:** ShengZhong Luo, Tianlong Wu.

**Investigation:** Tianlong Wu, Xigao Cheng.

**Methodology:** Tianlong Wu.

**Project administration:** Xigao Cheng.

**Supervision:** Tianlong Wu.

**Validation:** Xigao Cheng.

**Visualization:** Tianlong Wu.

**Writing – original draft:** ShengZhong Luo, Tianlong Wu, Xigao Cheng.

**Writing – review & editing:** ShengZhong Luo, Xigao Cheng.

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
