## [Decision Letter · Decision Letter 0]

23 Jul 2024

PONE-D-24-17982Spinal cord injury and risk of overall and type specific cardiovascular diseases: a meta-analysisPLOS ONE

Dear Dr. Luo,

Thank you for submitting your manuscript to PLOS ONE. After careful consideration, we feel that it has merit but does not fully meet PLOS ONE’s publication criteria as it currently stands. Therefore, we invite you to submit a revised version of the manuscript that addresses the points raised during the review process.

We look forward to receiving your revised manuscript.

Kind regards,

Amir Hossein Behnoush

Academic Editor

PLOS ONE

2. Please include a separate caption for each figure in your manuscript.

Additional Editor Comments (if provided):

Reviewers' comments:

Reviewer's Responses to Questions

**Comments to the Author**

1. Is the manuscript technically sound, and do the data support the conclusions?

Reviewer #1: Yes

Reviewer #2: Yes

2. Has the statistical analysis been performed appropriately and rigorously? 

Reviewer #1: Yes

Reviewer #2: Yes

3. Have the authors made all data underlying the findings in their manuscript fully available?

Reviewer #1: Yes

Reviewer #2: Yes

4. Is the manuscript presented in an intelligible fashion and written in standard English?

Reviewer #1: Yes

Reviewer #2: Yes

5. Review Comments to the Author

Reviewer #1: I appreciate the opportunity to read and review this interesting study.

The following article provides information about the risk of cardiovascular disease in patients with SCI.

Abstract

Well-written

Background:

Please cite if appropriate: https://www.sciencedirect.com/science/article/pii/S2214751923001664

Identify the complete writing form of GBD.

Some parts could be deleted to have a shorter and more effective introduction:

• “Higher incidence and prevalence rates …. access to quality healthcare globally.”

• Many individuals with SCI endure lasting disabilities, necessitating ongoing assistance for daily activities.

In the last sentence, please use past and passive voice.

Method

Identify the complete writing form of PROSPERO.

Results

Please add the number of studies from each countries after its name in parentheses (N=???).

Please add details of being prospective or retrospective and consider subgroup analysis for these two entities if possible.

Why Yoo 2024 is repeated twice in the figure?

Please specify each study’s RR/OR in the table.

Please inform us regarding what kind of comorbidities were talked in each study.

Discussion

No comment

Reviewer #2: Thank you for the opportunity to review the study examining the relationship between spinal cord injury (SCI) and cardiovascular diseases (CVDs). The authors have conducted a thorough exploration of the risks associated with various types of CVDs in the context of SCI, as well as the overall correlation between the two. To enhance the manuscript's quality for publication, I recommend addressing the following points:

General:

The punctuation throughout the text should be carefully reviewed and corrected where necessary.

Abstract:

Please include the date of your literature search in the methods subsection.

In the results subsection, specify the number of search results obtained.

It is unnecessary to mention the funnel plot and Egger's test in the methods subsection.

Introduction:

On page 4, the term "infraction" appears to be incorrect. Please review and correct this term.

Methods:

Clarify whether conference abstracts were excluded from the analysis.

Avoid using "etc." in the following sentence. Instead, list all the variables extracted: "The extracted data included details such as first author, publication date, country, event numbers, exposure numbers, confounders, etc."

Results:

Create a table detailing the studies excluded after full-text review, and provide the reasons for exclusion in the supplementary materials.

Given that fewer than 10 studies are included, the funnel plot's utility for assessing publication bias is limited. It is advisable to remove references to the funnel plot from both the methodology and results sections.

Discussion:

There are existing systematic reviews on the association between SCI and CVD risk factors. It would be beneficial to discuss the findings of these reviews in the discussion section.

Figures:

On page 6, the notation "=87.0%, Figure 5" is unclear. Please clarify what this refers to.

Ensure that all abbreviations used in the figures are defined in the figure captions.

By addressing these suggestions, the manuscript will be more coherent, comprehensive, and suitable for publication.

6. PLOS authors have the option to publish the peer review history of their article (what does this mean?). If published, this will include your full peer review and any attached files.

Reviewer #1: No

Reviewer #2: No

---

## [Author Response · Author response to Decision Letter 0]

25 Jul 2024

Response to reviewers

Dear editor and reviewers:

On behalf of my co-authors, we thank you very much for your comments on our manuscript entitled " Spinal cord injury and risk of overall and type specific cardiovascular diseases: a meta-analysis" (Manuscript ID: PONE-D-24-17982). 

We appreciate the constructive and valuable comments. We have revised our manuscript considerably according to your comments, questions, and suggestions. In the event that we missed any one of the comments please let us know. This document includes our responses to your comments point by point, and the revised portion are marked in RED in our manuscript. 

Reviewer 1

Comment 1:

Abstract

Well-written

Reply：Thank you for your positive and encouraging comments on our manuscript. Based on your insightful and constructive comments, we have carefully responded and revised the manuscript to improve its quality.

Comment 2: 

Background:

(1) Please cite if appropriate: https://www.sciencedirect.com/science/article/pii/S2214751923001664

Reply: Thank you for your advice. We did an in-depth study of this literature you provided, which is a pilot study of the combination of minocycline and methylprednisolone in the treatment of acute traumatic spinal cord injury. We tried everything to consider how to add it to the background section of our meta-analysis, unfortunately, we were all about describing the observational aspects without mentioning the appropriate interventions, so out of careful consideration we ended up not adding it to the background section, we look forward to your understanding and support.

(2) Identify the complete writing form of GBD.

Reply: Thank you for your careful advice. We have added the full name of GBD to the manuscript. (Page 3, Line 50 in red)

(3) Some parts could be deleted to have a shorter and more effective introduction:

• “Higher incidence and prevalence rates …. access to quality healthcare globally.”

• Many individuals with SCI endure lasting disabilities, necessitating ongoing assistance for daily activities.

Reply: Thank you for your valuable input, we removed the sentence you mentioned and the background section does read more concise and polished.

(4) In the last sentence, please use past and passive voice.

Reply：Thanks to your constructive comments, the tense of the last sentence was changed to passive and past tense.

Change in revised manuscript：

Page 4, Line 68~70 in red.

Therefore, a systematic review and meta-analysis of existing evidence from observational studies were conducted to quantify the overall and specific types of CVD risk in SCI.

Comment 3: 

Method

Identify the complete writing form of PROSPERO.

Reply：Thank you for your constructive comments. We have added the full name of PROSPERO.

Change in revised manuscript：

Page 4, Line 73~75 in red.

Our protocol has been registered on International prospective register of systematic reviews (PROSPERO) under the registration number (CRD42024537888).

Comment 4: 

Results

(1) Please add the number of studies from each countries after its name in parentheses (N=???).

Reply: Thanks to your constructive comments, we have added the number of studies included after the country name.

Change in revised manuscript:

Page 7, Line 147~149 in red.

The study populations predominantly originated from Taiwan Province in China (n=4), the United States (n=2), Canada (n=2) and Korea (n=1).

(2) Please add details of being prospective or retrospective and consider subgroup analysis for these two entities if possible.

Reply: Thank you for your in-depth review and constructive input. Indeed, when we designed this meta-analysis, we also considered whether it was possible to categorize the literature and analyze subgroups according to prospective cohorts or retrospective cohorts. However, after carefully reading the full text, we found that the data included in 7 study were data collected from public databases originating from various countries or regions, and the authors were only retrospectively analyzing the collected data, so these could be considered retrospective cohorts. Therefore, we have also mentioned this in the limitations section of the manuscript. We look forward to more high-quality, prospectively designed cohort studies at a later stage to explore the association between SCI and cardiovascular disease risk.

(3) Why Yoo 2024 is repeated twice in the figure?

Reply: Thank you for your careful review and kind questions. That's the case, because these articles provide effect sizes and confidence intervals for the different cardiovascular event outcomes, so when performing a meta-analysis of the risk of all-cause cardiovascular events, the different cardiovascular event outcomes in the same study can all be included in the analysis, and so there will be a situation in which you refer to the repetition of one study. We look forward to your further suggestions.

(4) Please specify each study’s RR/OR in the table.

Reply: Thank you for your valuable advice. We have added a column in Table 1 specifically to reflect the effect sizes of the included studies. Thank you for your kind and constructive reminder.

Change in revised manuscript:

Page 11, Column 13 of Table 1 in red.

(5) Please inform us regarding what kind of comorbidities were talked in each study.

Reply：Thank you for your pointed questions and comments. Admittedly, we did not consider comorbidities when we did this meta-analysis. At your suggestion, we went back and read the full article again and found that very few comorbidities mentioned in the literature were mainly pain, osteoporosis, etc., but these did not affect the analysis of the overall results, so we did not analyze them further in the revised manuscript, but thank you very much for reminding us to be aware of this point.

Comment 5: 

Discussion

No comment 

Reply：Thank you for the affirmation.

We really appreciate your positive and insightful suggestions on our manuscript. And we have revised our manuscript considerably according to your comments point by point. Under your kind help and professional guidance, we believe that our manuscript has been improved substantially. We are looking forward to your further suggestions.

Reviewer 2

Comment 1:

Thank you for the opportunity to review the study examining the relationship between spinal cord injury (SCI) and cardiovascular diseases (CVDs). The authors have conducted a thorough exploration of the risks associated with various types of CVDs in the context of SCI, as well as the overall correlation between the two. To enhance the manuscript's quality for publication, I recommend addressing the following points.

Reply：Thank you for your positive and encouraging comments on our manuscript. Based on your insightful and constructive comments, we have carefully responded and revised the manuscript to improve its quality.

Comment 2: 

General

The punctuation throughout the text should be carefully reviewed and corrected where necessary.

Reply: Thank you for your careful and kind reminder. We have re-examined the punctuation throughout the manuscript and made the appropriate corrections.

Comment 3: 

Abstract

(1) Please include the date of your literature search in the methods subsection.

Reply: Thank you for the kind reminder. We have added the date of retrieval to the methodology of the Abstract.

Change in revised manuscript：

Page 1, Line 18~19 in red.

The literature collection span is from database establishment until April 17, 2024.

(2) In the results subsection, specify the number of search results obtained.

Reply: Thank you for your kind reminder. We have added the number of literatures obtained.

Change in revised manuscript：

Page 2, Line 26 in red.

Our initial search generated 5357 relevant records form these international databases.

(2) It is unnecessary to mention the funnel plot and Egger's test in the methods subsection.

Reply：Thank you for kindly reminding us that we have removed the description of funnel plot and Egger test in the method.

Comment 4: 

Introduction

On page 4, the term "infraction" appears to be incorrect. Please review and correct this term.

Reply: Thank you for your kind reminder. The meaning expressed here is ' myocardial infarction ', which we have revised after consulting the dictionary.

Change in revised manuscript:

Page 4, Line 66 in red.

myocardial infarction

Comment 5: 

Methods

(1) Clarify whether conference abstracts were excluded from the analysis.

Reply: Thank you for kindly reminding us that the meeting summary will be excluded. We have included this in the exclusion criteria, and thank you very much for your careful review, which helped us better improve the quality of the manuscript.

(2) Avoid using "etc." in the following sentence. Instead, list all the variables extracted: "The extracted data included details such as first author, publication date, country, event numbers, exposure numbers, confounders, etc."

Reply：Thank you for your careful review and reminder. We have revised the description of extracting information to make it more complete.

Change in revised manuscript:

Page 5, Line 106 ~ 107 in red.

The extracted data included details such as first author, publication date, country, event numbers, exposure numbers, confounders, and effect size.

Comment 6: 

Results

(1) Create a table detailing the studies excluded after full-text review, and provide the reasons for exclusion in the supplementary materials.

Reply：Thank you for your careful and constructive feedback. We believe you are an expert in meta-analysis methodology and appreciate your guidance. We have created a literature exclusion table in the attachment, which specifies the titles and reasons for excluding articles as per your request.

Change in revised manuscript:

Supplementary Tables S4 Literature to be excluded after reading the full text

(2) Given that fewer than 10 studies are included, the funnel plot's utility for assessing publication bias is limited. It is advisable to remove references to the funnel plot from both the methodology and results sections.

Reply：Thank you for your kind reminder. We have removed the above content from the Methods and Results section.

Comment 7: 

Discussion:

There are existing systematic reviews on the association between SCI and CVD risk factors. It would be beneficial to discuss the findings of these reviews in the discussion section.

Reply：Thank you for your careful review and constructive feedback. This is a great suggestion. We also noticed an article titled 'A systematic review of cardiovascular risk factors in patients with traumatic spinal cord injury'. The reason why we did not compare it with him in the manuscript is as follows. First of all, the purpose of our two reviews is not the same thing. The main thrust of our manuscript was to directly explore the association between SCI and the risk of all-cause or specific types of cardiovascular events, and a quantitative Meta-analysis was conducted. However, it is clear that the systematic review in the one above is not talking about SCI and the risk of all-cause or specific cardiovascular disease, but is exploring which comorbidities or risk factors in patients with SCI, are risk factors for cardiovascular events. Second, our inclusion and exclusion criteria, where exposures and outcomes are not identical, are put together for comparison and do not highlight points of possible association in the 2 REVIEWS. Therefore, out of caution, we did not compare our results with previous reviews. We look forward to your understanding and support.

Comment 8: 

Figures:

(1) On page 6, the notation "=87.0%, Figure 5" is unclear. Please clarify what this refers to.

Reply：Thank you for your careful review and concern. The I-squared = 87% here is the magnitude of heterogeneity in forest plots for this subgroup of SCI patients with comorbidities.

(2) Ensure that all abbreviations used in the figures are defined in the figure captions.

Reply：Thank you for the kind and friendly reminder. We've annotated the full content of the acronyms involved below each Figure.

Comment 9: 

By addressing these suggestions, the manuscript will be more coherent, comprehensive, and suitable for publication.

Reply：

We really appreciate your positive and insightful suggestions on our manuscript. And we have revised our manuscript considerably according to your comments point by point. Under your kind help and professional guidance, we believe that our manuscript has been improved substantially. We are looking forward to your further suggestions.

Best wishes!

Yours sincerely,

Xigao Cheng 

E-mail: xigaocheng@hotmail.com

---

## [Decision Letter · Decision Letter 1]

13 Sep 2024

PONE-D-24-17982R1Spinal cord injury and risk of overall and type specific cardiovascular diseases: a meta-analysisPLOS ONE

Dear Dr. Luo,

Thank you for submitting your manuscript to PLOS ONE. After careful consideration, we feel that it has merit but does not fully meet PLOS ONE’s publication criteria as it currently stands. Therefore, we invite you to submit a revised version of the manuscript that addresses the points raised during the review process.

We look forward to receiving your revised manuscript.

Kind regards,

Amir Hossein Behnoush

Academic Editor

PLOS ONE

Journal Requirements:

Reviewers' comments:

Reviewer's Responses to Questions

**Comments to the Author**

1. If the authors have adequately addressed your comments raised in a previous round of review and you feel that this manuscript is now acceptable for publication, you may indicate that here to bypass the “Comments to the Author” section, enter your conflict of interest statement in the “Confidential to Editor” section, and submit your "Accept" recommendation.

Reviewer #2: All comments have been addressed

2. Is the manuscript technically sound, and do the data support the conclusions?

Reviewer #2: Yes

3. Has the statistical analysis been performed appropriately and rigorously? 

Reviewer #2: Yes

4. Have the authors made all data underlying the findings in their manuscript fully available?

Reviewer #2: Yes

5. Is the manuscript presented in an intelligible fashion and written in standard English?

Reviewer #2: Yes

6. Review Comments to the Author

Reviewer #2: I would like to thank the authors for addressing my comments.

-It is better to add citation to Table S4.

- Please cite Table S4 in the manuscript.

7. PLOS authors have the option to publish the peer review history of their article (what does this mean?). If published, this will include your full peer review and any attached files.

Reviewer #2: No

---

## [Author Response · Author response to Decision Letter 1]

16 Sep 2024

Dear editor and reviewers:

On behalf of my co-authors, we thank you very much for your comments on our manuscript entitled " Spinal cord injury and risk of overall and type specific cardiovascular diseases: a meta-analysis" (Manuscript ID: PONE-D-24-17982 _R1). 

We appreciate the constructive and valuable comments. We have revised our manuscript considerably according to your comments, questions, and suggestions. In the event that we missed any one of the comments please let us know. This document includes our responses to your comments point by point, and the revised portion are marked in RED in our manuscript. 

Reviewer 2

Comment 1: I would like to thank the authors for addressing my comments.

Reply：Thank you for your positive and encouraging comments on our manuscript. Based on your insightful and constructive comments, we have carefully responded and revised the manuscript to improve its quality.

Comment 2: 

-It is better to add citation to Table S4.

- Please cite Table S4 in the manuscript.

Reply: Thanks for your valuable suggestions, we have added the citations below the excluded literatures in Table S4, and at the same time added "the excluded literatures after reading the full text are in Table S4" (Page 7, Line 142~143 in red) in the revised manuscript. Thank you again for your meticulous review and advice.

Best wishes!

Yours sincerely,

Xigao Cheng 

E-mail: xigaocheng@hotmail.com

---

## [Editor Report · Decision Letter 2]

23 Sep 2024

Spinal cord injury and risk of overall and type specific cardiovascular diseases: a meta-analysis

PONE-D-24-17982R2

Dear Dr. Luo,

We’re pleased to inform you that your manuscript has been judged scientifically suitable for publication and will be formally accepted for publication once it meets all outstanding technical requirements.

Kind regards,

Amir Hossein Behnoush

Academic Editor

PLOS ONE
---

## [Editor Report · Acceptance letter]

17 Oct 2024

PONE-D-24-17982R2 

PLOS ONE

Dear Dr. Luo, 

I'm pleased to inform you that your manuscript has been deemed suitable for publication in PLOS ONE. Congratulations! Your manuscript is now being handed over to our production team.

Kind regards, 

on behalf of

Dr. Amir Hossein Behnoush 

Academic Editor

PLOS ONE